# Total Nitrite and Nitrate Concentration in Human Milk and Saliva during the First 60 Days Postpartum—A Pilot Study

**DOI:** 10.3390/biomedicines12061195

**Published:** 2024-05-28

**Authors:** Jenny Ericson, Michelle K. McGuire, Anna Svärd, Maria Hårdstedt

**Affiliations:** 1School of Education, Health and Social Studies, Dalarna University, 791 88 Falun, Sweden; 2Center for Clinical Research Dalarna, Uppsala University, 753 10 Uppsala, Sweden; anna.svard@regiondalarna.se (A.S.); maria.hardstedt@regiondalarna.se (M.H.); 3Department of Pediatrics, Falu Hospital, 791 82 Falun, Sweden; 4Margaret Ritchie School of Family and Consumer Sciences, University of Idaho, Moscow, ID 83844, USA; smcguire@uidaho.edu

**Keywords:** breastfeeding, human milk, nitrate, nitric oxide, nitrite, saliva

## Abstract

Nitric oxide (NO) in human milk may have important functions in lactation and infant health. This longitudinal pilot cohort study investigated the total nitrite and nitrate (NOx) concentration in human milk and maternal saliva during the first 60 days postpartum. Additionally, we explored the association between selected breastfeeding variables and milk and saliva NOx concentration. Human milk and maternal saliva samples were collected on days 2, 5, 14, 30, and 60 postpartum and analyzed for NOx concentration. Breastfeeding data were collected through self-assessed questions. Data analyses were performed using mixed models. The concentration of NOx in milk was significantly higher during the first 30 days compared to day 60, and there was a positive association between milk and saliva NOx concentrations throughout the entire study period. In absolute numbers, partially breastfeeding mothers had a lower concentration of NOx in milk on day 2 compared to exclusively breastfeeding mothers (8 vs. 15.1 μM, respectively). Partially breastfeeding mothers reported a later start of secretory activation and fewer mothers in this group started breastfeeding within the first hour after birth. Due to the small numbers, these differences could not be statistically evaluated. Further research is warranted to elucidate the role of NO in lactation success and breastfeeding outcomes.

## 1. Introduction

Breastfeeding provides health benefits for both mothers and infants and reduces healthcare costs even in high-income countries [1]. Human milk fulfills the nutritional needs of the infant, regulates the immune system, protects against pathogens, and promotes neurological and metabolic development. Breastfeeding also offers long-term health advantages for the mother [1,2,3,4]. The World Health Organization recommends initiating breastfeeding within the first hour after birth and exclusively breastfeeding for the first six months, followed by continued partial breastfeeding for up to two years or more [5]. However, many mothers cease breastfeeding during the first months, often earlier than intended [6,7], and the underlying causes for this are inadequately understood. Previous studies have shown that the most common reasons for breastfeeding cessation are breastfeeding difficulties, including perceived low milk production [7,8]. As 70% of mothers experience breastfeeding difficulties, a statistic that is associated with higher rates of non-exclusive breastfeeding and subsequent increased risk of breastfeeding cessation, this is an important issue to address [7,8].

Nitric oxide (NO) and its metabolites serve as essential mediators in various physiological processes in the human body including neurotransmission, the regulation of vascular tone, host defense, and cellular respiration [9]. Nitrite and nitrate (referred to collectively, herein, as NOx), which are both relatively high in concentration in human milk [10], are believed to be involved in the physiological adaptation of infants to extrauterine life and may be involved in the process of lactation [11]. During the neonatal period, the infant has an immature immune system, less acidic gastric juices, and reduced bacterial communities in the mouth, which are all associated with increased susceptibility to infections. It is plausible that human milk-borne nitrite may contribute to antimicrobial activity in the infant’s gastrointestinal tract [11]. In addition, several studies have indicated that NO may be involved in milk production and/or letdown [11,12,13].

In adults, the entero-salivary nitrate–nitrite–nitric oxide pathway contributes to a stable nitrite pool in the blood by ensuring an active uptake of dietary nitrate from the circulation by salivary glands and, subsequently, an incorporation of nitrate into saliva [11]. Bacteria in the mouth reduce salivary nitrate to nitrite, which is further reduced to NO and other nitrogen-containing metabolites in the stomach. Nitrite in human milk may compensate for the immature entero-salivary nitrate–nitrite–nitric oxide pathway of the infant. Human milk is an important source of nitrite, which may be relevant for both the formation of NO metabolites in the stomach of the infant as well as NO bioavailability in the systemic circulation [10,14].

The source of NOx in human milk remains unclear, but concentrations in milk appear to be independent of maternal nitrate intake [11]. Previous data suggest that higher nitrite concentrations in milk during the first days after birth are associated with an earlier lactation onset and greater milk output [15,16]. However, associations between concentration of NOx in milk and saliva and breastfeeding outcomes have been underexplored, and very little research has documented NOx concentration in human milk over time. Understanding the NO system and its regulation in breastfeeding women might be a key to understand both breastfeeding success as well as difficulties.

Hence, the primary aim of this pilot study was to document the total NOx concentration in human milk and maternal saliva collected over the first 60 days postpartum. In addition, we explored associations between NOx concentrations in milk and breastfeeding characteristics/outcomes such as exclusive versus partial breastfeeding.

## 2. Materials and Methods

### 2.1. Study Design and Participants

Data presented here were collected as part of a longitudinal cohort pilot study designed to understand factors related to lactation success in Swedish women. From 2021 to 2022, pregnant women in the county of Dalarna, Sweden were recruited through social media, leaflets, and/or midwives. Using a consecutive recruitment method, pregnant women (≥18 years) who could answer questionnaires in Swedish and who consented to participate were included in the study.

### 2.2. Milk, Saliva, and Data Collection

Breastfeeding registration and self-report questionnaires regarding breastfeeding were collected through a mobile application on days 2, 5, 14, 30, and 60 postpartum or until breastfeeding cessation. Exclusive breastfeeding was defined as providing only human milk to the infant whereas partial breastfeeding involved providing a combination of human milk and infant formula.

The mothers provided five 3 mL samples of milk from one breast (not specified which) and three 2 mL samples of saliva on days 2, 5, 14, 30, and 60 postpartum or until breastfeeding cessation. Milk was collected by the mothers in their homes, in the morning, using a conventional manual breast pump provided by the research project. Women were not required to provide complete breast expressions, so milk was generally foremilk. Saliva samples were collected via passive drooling, where mothers allowed saliva to drip naturally into the tubes. Polypropene cryotubes (Sarstedt AG & Co. KG, Nümbrecht, Germany) were used for the saliva, and polystyrene tubes (Sarstedt AG & Co. KG, Nümbrecht, Germany) were used for the milk. Milk and saliva were frozen immediately at home and kept in home freezers (−20 °C) for up to 60 days, when they were transferred to a −80 °C freezer at the hospital.

### 2.3. Analysis

After thawing, milk was centrifuged three times to separate the aqueous phase from the lipid layer and cells (pellet): twice at 680× *g* (10 min, 4 °C) and once at 10,000× *g* (30 min, 4 °C) [17]. Saliva was thawed and centrifuged at 10,000× *g* for 10 min to separate solid particles (pellet) from the liquid portion (supernatant). Following centrifugation, supernatants from milk and saliva preparations were transferred to 96-well plates and frozen at −80 °C until further analysis. Total concentrations of NOx were measured using the Cayman’s nitrite/nitrate colorimetric assay kit (no 780001) according to the manufacturer’s instructions [18]. The Cayman colorimetric kit has previously been used for human milk [12]. The intra-assay % coefficients of variability (% CV) were 4.1 (SD 5.1) for breastmilk and 8.8 (SD 12.0) for saliva. Inter-assay % CV based on breastmilk analysis was 13.9 (SD 2.0).

Analyses of somatic cell count (SCC) and levels of sodium (Na) and potassium (K) were performed on whole milk according to the manufacturer’s instructions. Somatic cell count was analyzed with DeLaval cell counter (DeLaval International AB, Tumba, Sweden). Sodium and potassium levels were determined using ion selective electrodes (sodium: LAQUAtwin Na-11; potassium: LAQUAtwin K-11; Horiba, Japan). The ratio between sodium and potassium was calculated; a ratio above 0.8 indicated mastitis.

### 2.4. Statistical Analyses

Data analyses were conducted using IBM SPSS Statistics for Windows (version 28.0). Descriptive statistics are presented here as medians and interquartile ranges (IQRs) or numbers and percentages. Total NOx concentrations in milk and saliva, together with breastfeeding characteristics, have been presented in total (all women) and subdivided by breastfeeding status, and exclusive versus partial breastfeeding, at day 60 postpartum. No analyses of comparative statistics were performed between mothers who exclusively versus partially breastfeed due to few cases in the latter group.

All statistics were performed on log-transformed data due to a right-skewed data distribution. Linear mixed-effect models were used to analyze the trajectory (i.e., days 2, 5, 14, 30, 60 postpartum) of NOx concentrations in milk and saliva. A linear mixed-effect model was also used to evaluate associations between NOx concentrations in milk and saliva. The linear mixed-effects modeling included repeated measurements with fixed effects of time. We used the covariance structure AR1, and model assumptions were checked. Reference levels (coded as 1) were taken at day 60 postpartum. If the value for the estimate was positive, the NOx concentration was lower in the reference group. If the value for the estimate was negative, NOx concentration was higher in the reference group. The linear mixed-effects model offers an advantage in dealing with missing outcome values as it allows for the inclusion of individuals in the analysis even when some outcome values are not available. Furthermore, it is a valuable statistical tool for analyzing longitudinal data due to its ability to accommodate the dependence among repeated measurements from the same individual over time. The results from the linear mixed-effects model analyses are presented here with estimates, which should be interpreted as the mean differences in total NOx concentration, 95% confidence intervals (95% CI), and *p*-values. Differences were considered significant if *p* < 0.05.

## 3. Results

Altogether, 25 mothers provided milk and saliva samples. Background characteristics and demographics of the participating mothers and their infants are presented in Table 1. Not all mothers could provide samples at all time points, and the total numbers of samples collected are presented in Table 2.

### NOx Concentration in Milk and Saliva

The median total NOx concentration in milk collected on days 2, 5, 14, 30, and 60 postpartum was 12.3 μM (range 3.0–54.4 μM), and in saliva, it was 53.6 μM (range 0.2–354.3 μM) (Table 2 and Figure 1). The NOx concentration in milk was higher on days 2, 5, 14, and 30 postpartum compared to day 60 (estimate 0.88 (95% CI 0.76–1.00), *p* < 0.001) (Table 3). In saliva, total NOx concentration was lower on day 2 compared to days 5, 14, 30, 60 (estimate 1.72 (95% CI 1.42–2.01), *p* < 0.001) (Table 3). Based on the mixed model analysis, there was an association between NOx concentrations in milk and saliva (estimate 0.16 (95% CI 0.06–0.27), *p* = 0.020, result not shown in the table).

NOx concentration in milk during the first days postpartum was lower in partially breastfeeding mothers compared to exclusively breastfeeding mothers: 8 μM (range 3.2–22.2) vs. 15.1 μM (range 3.8–54.4), respectively. Partially breastfeeding mothers also reported a longer period of time until secretory activation (also referred to as secretory activation or “milk coming in”): 84 vs. 72 h compared to mothers still exclusively breastfeeding on day 60 postpartum. Furthermore, fewer of the partially breastfeeding mothers had initiated breastfeeding during the first hour after birth compared to mothers still exclusively breastfeeding on day 60 postpartum [1 (33%) vs. 16 (80%)]. However, these differences could not be statistically evaluated due to a small sample size. (Table 2). For two mothers, one at day 14 and one at day 30, self-reported symptoms, a high somatic cell count (1709 and 352 cells/µL) and Na/K ratio (1.04 and 1.24), suggested mastitis at the time of sampling. The total NOx concentrations in milk for these two mothers (16.9 and 15.2 µM) were slightly higher than the median concentrations of 14.4 µM on day 14 and 12 µM on day 30 based on all mothers.

## 4. Discussion

In summary, we have presented the natural time course of the total NOx concentration in human milk and maternal saliva during the first 60 days postpartum. The NOx concentration was higher on day 2 than on day 60, which was in line with the few previous studies published [14,15]. The NOx concentration remained relatively high until day 30, which means that not only colostrum, but also transitional and mature milk during the first month have high concentrations of NOx.

The concentration of NOx was higher in saliva than in milk, as previously shown [19]. A recent study examining NOx in plasma, saliva, and milk at one time point, at least 120 days after birth, found no association between nitrite concentrations in plasma and nitrite concentrations in milk. The authors suggested that complex mechanisms regulate concentrations of nitrite and nitrate in milk, probably reflecting the fact that NO is synthesized by the L-arginine–NO-synthase pathway locally in the mammary gland [19]. This was supported by data from Iizuka et al. [15] who did not find associations between NOx in plasma and milk. Interestingly, we found an association between NOx concentration in milk and saliva, based on repeated samples over a period of 60 days postpartum. Due to our small sample size, however, these results need to be further evaluated. In this study, we did not collect plasma samples, which was a limitation. Larger studies analyzing NOx, or nitrate and nitrite, simultaneously in milk, saliva, and plasma are needed to explore the association between concentrations in different body fluids in breastfeeding mothers.

Previous studies have suggested that NO may trigger lactation in humans with NOx concentration in milk peaking just before the initial increase in milk volume [16]. Furthermore, the NOx concentration was overall higher in milk produced by mothers characterized as high-milk-secreting compared to those producing lesser amounts of milk during the first days after birth [16]. In the present study, almost all women breastfed exclusively. Thus, we could not fully evaluate the potential association between NOx concentration in milk and breastfeeding exclusivity. However, we noted that mothers who would be only breastfeeding on day 60 postpartum (compared to those who would be exclusively breastfeeding on day 60 postpartum) tended to produce milk with lower NOx concentration at day 2 postpartum. These women also experienced delayed secretory activation, and fewer of them initiated breastfeeding during the first hour after birth compared to exclusively breastfeeding mothers. These differences could, however, not be statistically evaluated due to a small sample size. Only two mothers reported to have experienced mastitis during the study period; hence, our sample size was too small to assess any potential association between mastitis and NOx concentration. Interestingly, in cows and goats, mastitis is linked to increased concentrations of nitric oxide, nitrite, and nitrate [20,21]. Nitric oxide is believed to play a crucial role in the immune system, including antimicrobial effects [22]. Nevertheless, future studies should explore the relationship between mastitis and NO levels in humans as it appears to vary across different species [20].

There were several limitations to this study. First, the sample size was small, and the results should be interpreted with this in mind. Moreover, the duration of the study was only 60 days, and a longer follow-up could possibly allow more insight as to whether early NOx concentrations in milk might predict more long-term breastfeeding outcomes. Not all mothers provided samples at all occasions. However, the statistical mixed model method has the advantage of handling missing data. Another limitation was that the mothers represented a quite homogenous population, having high education levels, being multiparous, and showing a high percentage of exclusive breastfeeding. Previous studies have shown that highly educated women tend to breastfeed more exclusively and for a longer time [1]. For future research, the ability to recruit a broader population is important.

Nonetheless, no studies have documented NOx in milk and maternal saliva as extensively and longitudinally as the current study. Moreover, few studies have explored the physiological mechanisms of breastfeeding success and difficulties, and even fewer have focused on nitrite and nitrate concentration. This pilot study is part of a comprehensive research study with the goal of exploring the physiological causes of breastfeeding difficulties.

## 5. Conclusions

Based on the presented pilot study, the total concentration of NOx in human milk was higher during the first 30 days compared to day 60 postpartum. An association between NOx concentration in milk and saliva was found over the study period of 60 days postpartum. Since NO and its metabolites are believed to be essential for lactation, besides the suggested importance for the infant, studying NOx can provide clues both to the physiology of successful breastfeeding as well as to breastfeeding difficulties. Further larger longitudinal studies are needed.

## Figures and Tables

**Figure 1 biomedicines-12-01195-f001:**
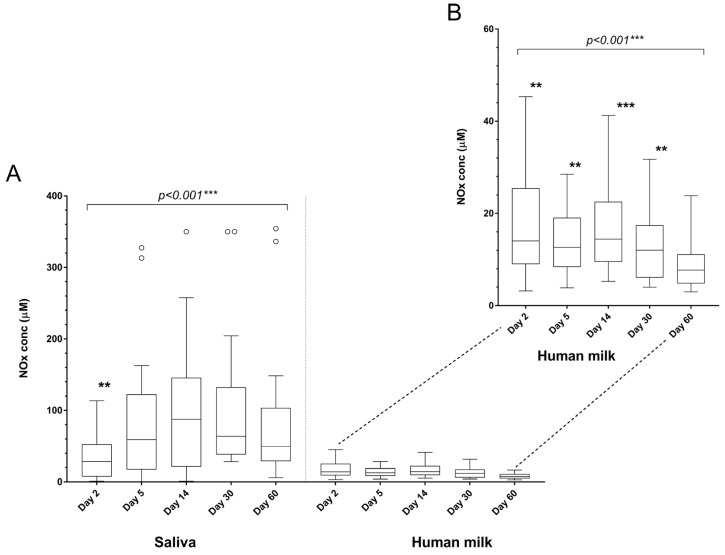
(**A**). Total nitrate and nitrite (NOx) concentration (μM) in milk and saliva on days 2, 5, 14, 30, and 60 postpartum. (**B**). Graph adjusted to better visualize NOx concentration in milk. Data are presented as Tukey boxplots; the box extends from the 25th to the 75th percentile with a horizontal line for the median; whiskers show maximum and minimum values, and outliers (>1.5* interquartile range) are plotted as individual dots. Presented in the figure are significant levels for the overall mixed model analysis and estimated marginal means compared to the concentration at day 60 (** *p* < 0.01, *** *p* < 0.001).

**Table 1 biomedicines-12-01195-t001:** Background demographics of participating mothers and infants (*n* = 25) and data related to pregnancy, delivery, and breastfeeding. Data are presented as medians and interquartile ranges [IQRs] or numbers and percentages (%).

	Total
Maternal demographics and health	
Age at time of birth, y [IQR]	32 [32–35.5]
Completed university education, *n* (%)	20 (80)
Cohabiting at time of birth, *n* (%)	21 (88)
Body mass index, kg/m^2^ [IQR]	24.2 [21.5–26.9]
Maternal chronic disease *, *n* (%)	3 (12)
Mental illness ^#^, *n* (%)	7 (29)
Diabetes (type 1, type 2, or gestational diabetes), n (%)	3 (13)

Pregnancy, delivery, and breastfeeding characteristics	
Primiparous, *n* (%)	7 (30)
Pregnancy complication, *n* (%)	2 (7)
Cesarean section, n (%)	2 (8)
Earlier breastfeeding experience, n (%)	15 (60)
Exclusive breastfeeding at discharge from maternity unit, *n* (%)	24 (100)

Infant characteristics	
Gestational age at birth, wk [IQR]	39 [39,40]
Male, *n* (%)	15 (63)
Birth weight, g [IQR]	3599 [3296–3974]

* Two mothers had asthma, one had a non-specified endocrine decease, and one had psoriasis arthritis. ^#^ The kind of mental illness was not specified.

**Table 2 biomedicines-12-01195-t002:** Total nitrate and nitrite (NOx) concentrations (μM) in milk and saliva and data related to breastfeeding in the 25 participating mothers up to 60 days postpartum. Data are presented as medians and intra-quartile ranges [IQR] or numbers and percentages (%), respectively. Data are given for all women as well as subdivided by breastfeeding status at day 60 postpartum.

		All Women		Exclusive Breastfeeding on Day 60 Postpartum		Partial Breastfeeding on Day 60 Postpartum
	N	median [IQR] or number (%)	*n*	median [IQR] or number (%)	*n*	median [IQR] or number (%)
NOx (μM) in milk						
day 2	17	14.0 [3.2–54.4]	14	15.1 [3.8–54.4]	3	8.0 [3.2–22.2]
day 5	20	12.7 [3.8–28.5]	18	12.7 [3.8–28.5]	2	11.3 [7.5–15.1]
day 14	18	14.4 [5.3–41.3]	15	14.0 [5.3–38.0]	3	23.2 [13.3–41.3]
day 30	22	12 [4.0–48.5]	18	12.6 [4.0–48.5]	4	9.3 [5.7–17.1]
day 60	22	7.7 [3.0–23.8]	18	7.7 [3.0–23.8]	4	8.1 [3.4–13.8]
NOx (μM) in saliva						
day 2	12	28.6 [1.0–113.5]	11	17.5 [1.0–113.5]	1	59.3 [59.3]
day 5	16	59.1 [0.2–327.4]	13	51.4 [0.2–313.1]	3	104.8 [78.3–327.4]
day 14	16	87.6 [1.0–350]	13	82.6 [1.0–171.9]	3	257.4 [84.6–350.0]
day 30	18	63.7 [28.3–350]	15	59.3 [28.3–350.0]	3	87.5 [47.4–350.0]
day 60	17	49.2 [5.9–354.3]	13	43.0 [5.9–354.3]	4	103.7 [28.2–336.3]
Breastfeeding characteristics						
Breastfed during the first hour after birth	23	17 (74)	20	16 (80)	3	1 (33)
Reported secretory activation hours after birth	21	72 [24–120]	19	72 [48–120]	2	84 [72–96]
Exclusive breastfeeding day 2	24	22 (92)	21	21 (100)	3	1 (33)
day 5	23	20 (87)	20	19 (95)	3	1 (33)
day 14	23	21 (88)	19	18 (95)	4	1 (25)
day 30	23	19 (86)	19	18 (95)	3	1 (33)
day 60	23	21 (84)	21	21 (100)		
Number of breastfeeding sessions per day day 2	14	11 [7]	12	10.5 [5.8]	2	15
day 5	15	11 [7]	13	11 [7.5]	2	10
day 14	15	10 [3]	14	8 [2.8]	1	7 [0]
day 30	14	10 [5]	12	10.5 [5.5]	2	9
day 60	15	9 [6]	13	11 [4]	2	13

**Table 3 biomedicines-12-01195-t003:** Results of mixed model analysis comparing total concentrations of nitrite/nitrate (NOx, μM) in milk and saliva on days 2, 5, 14, 30, and 60 postpartum. Data are presented with estimates, 95% confidence intervals (95% CI), and *p*-values.

		Human Milk			Saliva	
	n	Estimate (95% CI)	*p*-Value	n	Estimate (95% CI)	*p*-Value
Model	25	0.88 (0.76–1.00)	<0.001	20	1.72 (1.42–2.01)	<0.001
day 2		0.27 (0.10–0.44)	0.002		−0.52 (−0.96–0.09)	0.019
day 5		0.22 (0.60–0.38)	0.008		−0.16 (−0.52–0.23)	0.42
day 14		0.29 (0.14–0.44)	<0.001		−0.005 (−0.37–0.36)	0.98
day 30		0.18 (0.05–0.30)	0.005		0.16 (−0.13–0.46)	0.28
day 60		ref			ref	

## Data Availability

The dataset generated in the current study is not publicly available due to ethical and legal reasons but is available from the corresponding author on reasonable request.

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
