# Peer review of "Total Nitrite and Nitrate Concentration in Human Milk and Saliva during the First 60 Days Postpartum—A Pilot Study"

_biomedicines, 2024, doi:10.3390/biomedicines12061195_

Round 1

Reviewer 1 Report

Comments and Suggestions for Authors

The work reports a a pilot study focused on combined nitrite and nitrate concentration in human milk and saliva during the first 60 days postpartum. Human milk and saliva samples were analyzed for nitrite/nitrate concentration at 5 days postpartum. It was demonstrated that nitrite and nitrate concentration was significantly higher in milk during the first 30 days compared to day 60.  

Despite interesting, I believe that the present work does not reach the high-quality standards of the Biomedicines journal. The results presented are only preliminary and the way the authors evaluated the total concentration of combined nitrite/nitrate is not accurate (Cayman's nitrite/nitrate colorimetric assay!!!)

English is rather poor, and an extensive polishing is required. I invite the authors to use impersonal form throughout the whole manuscript.

Section 2, line 76. The authors should specify what they mean for “mixed-method approaches”.

Section 3. Results. I consider the exclusion of 7 potential samples not important for the present study. Thus, I would only report the recruitment of 25 samples (Figure 1 must be erased). On the other hand, the rationale for choice of the 25 samples, reported in Table 2, that should be renamed as Table 1, should be critically discussed.

Analysis of combined nitrite/nitrate, line 102. The authors should specify if they adhere to the same protocol reported by Kverka et al., in Clin Chem, 2007.

5. Conclusion. This section is too maigre. It should be expanded by focusing on the main results achieved and future perspectives.

Comments on the Quality of English Language

Extensive revision

Author Response

Thank you for your effort to review our manuscript and for your valuable comments. Changes are marked in yellow in the revised manuscript, but it is noteworthy that text that we removed were not retained as strikethrough text. Please find below our replies to the comments.

Reviewer 1

Despite interesting, I believe that the present work does not reach the high-quality standards of the Biomedicines journal. The results presented are only preliminary and the way the authors evaluated the total concentration of combined nitrite/nitrate is not accurate (Cayman's nitrite/nitrate colorimetric assay!!!)

Reply: Thank you for the remark. As you suggest, these results are from a pilot study. Nevertheless, analysis of nitric oxide metabolites in human milk, despite an obvious physiological and clinical interest, are scarce in the literature.

We respectfully disagree with the comment related to our laboratory methods, which have been used by others for this purpose (please see above). Although there are likely more rigorous methods of analysis that could be used in the future, our study provides a preliminary look at NOx concentrations in milk and maternal saliva over the first 60 days postpartum. We believe that our findings, although limited, provide new and important information for the field of human milk and lactation.

English is rather poor, and an extensive polishing is required. I invite the authors to use impersonal form throughout the whole manuscript.

Reply: Thank you for pointing this out. One of your coauthors (MKM) speaks English as a first language and has thoroughly evaluated the language prior to resubmission. However, if the editors would like for us to have a professional language review, we can certainly do that. Changes not marked in manuscript.

Section 2, line 76. The authors should specify what they mean for “mixed-method approaches”.

Reply: Thank you for noticing that. We have rephrased the sentence and removed mixed methos approaches as this is not relevant for this study.

Section 3. Results. I consider the exclusion of 7 potential samples not important for the present study. Thus, I would only report the recruitment of 25 samples (Figure 1 must be erased). On the other hand, the rationale for choice of the 25 samples, reported in Table 2, that should be renamed as Table 1, should be critically discussed.

Reply: Thank you for this comment. We have removed figure 1 and added text in the discussion regarding limitations, stating that not all mothers provided samples at all occasions, which may have affected the results. This study was designed to be pragmatic. That is, inclusive to mothers also having breastfeeding difficulties resulting in not being able to provide samples at all the time points. The aim is to follow mothers both with functioning and non-functional breastfeeding and to be able to keep as many mothers as possible in the study over the follow-up period.

Analysis of combined nitrite/nitrate, line 102. The authors should specify if they adhere to the same protocol reported by Kverka et al., in Clin Chem, 2007.

Reply: We did not adhere to the same protocol as in this protocol. Based on our comment, we changed the reference to a paper using the same kit (Caymans) as we did (Ohta et al. 2004).

  1. Conclusion. This section is too maigre. It should be expanded by focusing on the main results achieved and future perspectives.

Reply: We agree that it may be too thin. We have added a few sentences on relevant findings.

Reviewer 2 Report

Comments and Suggestions for Authors

A very interesting article written by experts in the field based on a preliminary data from a solid study.

Specific comments:

1.      The data are valuable and interesting but preliminary and limited in their content. Thus, this work must be reclassified from “Article” to “communication”.

2.      In the abstract, it is unclear from whom the saliva samples were obtained (i.e. mothers).

3.      A graphical abstract, even more technical (a study design) than mechanistic could be considered.

Author Response

Thank you for your effort to review our manuscript and for your valuable comments. Changes are marked in yellow in the revised manuscript, but it is noteworthy that text that we removed were not retained as strikethrough text. Please find below our replies to the comments.

Reviewer 2

A very interesting article written by experts in the field based on a preliminary data from a solid study.

Reply: We are grateful for your encouraging comments.

Specific comments:

The data are valuable and interesting but preliminary and limited in their content. Thus, this work must be reclassified from “Article” to “communication”.

Reply: We are happy to comply with this request. We have discussed this option in the research group. If the text could stay unchanged in length, we welcome a change to a communication on the Editors request. Since the data is based on a pilot study, this format might be more suitable.

In the abstract, it is unclear from whom the saliva samples were obtained (i.e. mothers).

Reply: Thank you for your comment, we have added “maternal” before saliva to clarify from whom the samples were obtained.

A graphical abstract, even more technical (a study design) than mechanistic could be considered.

Reply: Thank you for this suggestion. A graphical abstract is not mandatory for the publication. If the Editor would like us to add a graphical abstract, we are happy to work on this.

Round 2

Reviewer 1 Report

Comments and Suggestions for Authors

The authors have adequately addressed all remarks. I am still of the idea that the findings are limited; however, it can bring slightly additional information in the field of human milk and lactation.

Reviewer 2 Report

Comments and Suggestions for Authors

None